# Personalised Distillation: Empowering Open-Sourced LLMs with Adaptive Learning for Code Generation

**Hailin Chen**[*♣♠], **Amrita Saha**[*♠], **Steven HOI**[♠], **Shafiq Joty**[♣♠]

♣ Nanyang Technological University, Singapore

♠ Salesforce Research

{hailin001, srjoty}@ntu.edu.sg
{amrita.saha, shoi}@salesforce.com

## Abstract

With the rise of powerful closed-sourced LLMs (ChatGPT, GPT-4), there are increasing interests in distilling the capabilities of close-sourced LLMs to smaller open-sourced LLMs. Previous distillation methods usually prompt ChatGPT to generate a set of instructions and answers, for the student model to learn. However, such standard distillation approach neglects the merits and conditions of the student model. Inspired by modern teaching principles, we design a personalised distillation process, in which the student attempts to solve a task first, then the teacher provides an adaptive refinement for the student to improve. Instead of feeding the student with teacher's prior, personalised distillation enables personalised learning for the student model, as it only learns on examples it makes mistakes upon and learns to improve its own solution. On code generation, personalised distillation consistently outperforms standard distillation with only one third of the data. With only 2.5-3K personalised examples that incur a data-collection cost of 4-6$, we boost CodeGen-mono-16B by 7% to achieve 36.4% pass@1 and StarCoder by 12.2% to achieve 45.8% pass@1 on HumanEval.[1]

## 1 Introduction

Recently, powerful close-sourced large langauge models (LLMs) including ChatGPT, GPT-4 have become predominant, accumulating over 170 million users within 5 month of its launch. Such close-sourced LLMs demonstrate strong performance in a wide range of tasks, from improving writing proficiency to code generation. However, due to their closed-source nature, concerns have been raised regarding factors such as the availability of these services, high associated costs, concerns on ethics and safety, and potential data privacy implications, all of which limit their seamless integration into real-world applications. In light of these concerns, a natural question arises: Can we distill the remarkable abilities exhibited by closed-source LLMs into smaller open-source LLMs?

Researchers have explored such distillation idea (Taori et al., 2023; Wang et al., 2022; Xu et al., 2023b), by querying ChatGPT to generate task instruction and solution pairs, and using the collected data to finetune a student model. However, this standard distillation approach fits different student models to the same data distribution (teacher's prior), disregarding their unique abilities and capacity. In education domain, personalised learning which provides customized learning experience that adapts to student's learning progress and capacity, has proven highly effective and widely adopted (Roberts-Mahoney et al., 2016; Shemshack and Spector, 2020). Inspired by such finding, we hypothesize that personalised learning is also beneficial for model distillation.

In this work, we propose personalised distillation and empirically evaluate its effectiveness in the domain of code generation. Similar to standard distillation, we first employ ChatGPT to generate task instructions accompanied by unit test cases. Then we follow three steps for personalized distillation as shown in Figure 1. First, we let the student model attempt to solve the task. Then, we evaluate the student's attempt with unit test cases and get execution feedback. If the execution feedback contains errors, in the final step we prompt the teacher model (ChatGPT) to refine the student's attempt.

Such data collection process makes the learning experience both interactive — as the student participates to make attempts, and personalised — both the input (tasks) and output (refinement data) are customised to the student. Essentially, personalised labeled data help the student to refine its own policy, rather than adopting a new prior of the teacher.

With the personalized code data as target out-

---

*These authors contributed equally to this work

[1]Our codes will be available at https://github.com/salesforce/PersDistill

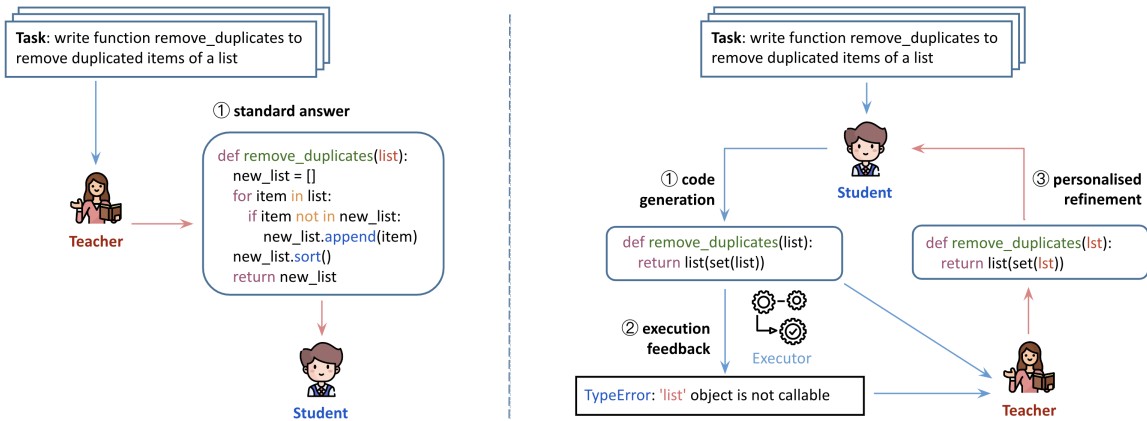

Figure 1: Overview of our framework. **Left: standard distillation**. ① Teacher generates standard answer to a given problem for the student to learn **Right: personalised distillation**. ① Student first generates its own attempt to solve the task. ② Executor evaluates generated code with unit test cases. ③ Teacher provides adaptive refinement given student's attempt and its execution feedback.

put, we construct three variants of finetuning data (i) PERsD data which formats it as a typical text-to-code generation task, (ii) PERsD-refine which treats it as a code-refinement task, given a task instruction, incorrect code and execution error feedback (ii) PERsD-combine which simply combines PERsD and PERsD-refine finetuning data, i.e. code generation and refinement tasks.

We collect 10K standard distillation examples and around 2.5-3K personalised examples for pretraining. Through zero-shot evaluation on HumanEval (Chen et al., 2021) and MBPP (Austin et al., 2021), we observe that all PERsD variants consistently outperform their counterparts which use standard distillation. This compelling result strongly validates our hypothesis regarding the advantages of personalized distillation. Ablation studies further reinforce our hypothesis, uncovering intriguing properties such as the benefits of multi-round personalized distillation and the ability of our models to leverage execution feedback for self-correction. Notably, personalised distillation boosts the state-of-the-art open-sourced pretrain model StarCoder (Li et al., 2023a) significantly — by 12.2% to achieve 45.8 in pass@1 and 82.3 in pass@100 on HumanEval.

## 2 Related Work

### 2.1 Distillation from ChatGPT

Previous works have explored distillation from ChatGPT including Alpaca(Taori et al., 2023), Vicuna(Chiang et al., 2023) and Baize(Xu et al., 2023b). However, these works can all be considered as standard distillation as they do not consider the conditions and capacity of student model. Wiz-

| Methods | Personalised | Interactive | Code-related |
|---|---|---|---|
| Alpaca | ✗ | ✗ | ✗ |
| Vicuna | ✗ | ✗ | ✗ |
| Baize | ✗ | ✗ | ✗ |
| WizardLM | ✗ | ✗ | ✗ |
| WizardCoder | ✗ | ✗ | ✓ |
| Lion | Input | ✓ | ✗ |
| PERsD | Input + Output | ✓ | ✓ |

Table 1: Related work on distillation from ChatGPT

ardLM(Xu et al., 2023a) and WizardCoder(Luo et al., 2023) iteratively prompts teacher model to generate more complex instructions. Their approach can be seen as an orthogonal advancement that can potentially be combined with personalised distillation.

Lion (Jiang et al., 2023) proposes to incorporate student model's answer and sample more hard tasks for which the student failed to solve. Thus, Lion can be considered as input personalised distillation as only the input tasks are customised for different student. Our approach differs as we provide customization both on input and output, and we empirically show that personalising labels is critically beneficial.

### 2.2 Code Generation with Feedback

Recently, there has been an increasing amount of research on exploring on how to use feedback for an iterative and improved code generation through code-refinement. Self-refine(Madaan et al., 2023), Self-debug(Chen et al., 2023b) and Reflexion (Shinn et al., 2023) are inference-time methods which use powerful close-sourced LLMs to generate better code from internal or external feedback. Although they show high performance,

| Methods | Training | | | Inference | |
| --- | --- | --- | --- | --- | --- |
| | Single Model | Data Source | Personalised | w/ execution feedback | w/o ChatGPT |
| Self-refine | ✓ | No Training | ✗ | ✗ | ✗ |
| Self-debug | ✓ | No Training | ✗ | ✓ | ✗ |
| Reflexion | ✓ | No Training | ✗ | ✓ | ✗ |
| Self-edit | ✗ | Standard GT | ✗ | ✓ | ✓ |
| Self-correct | ✗ | Self-exploration | ✓ | ✓ | ✓ |
| ILF | ✗ | Human labeled | ✓ | ✓ | ✓ |
| PERsD-refine | ✓ | ChatGPT | ✓ | ✓ | ✓ |

Table 2: Related work on Code Generation w/ feedback

these methods are limited as they require access to close-sourced LLMs.

Self-edit (Zhang et al., 2023) trains a separate code editor to rectify generated code from a base LLM. The training label is from original gold answer, thus not label-personalised. Similarly, Self-correct (Welleck et al., 2022) trains a separate corrector model to rectify the output from a fixed generator model. However, the training label is from self-exploration of the corrector model: sampling multiple refinements and choosing the one leading to higher reward. Finally, ILF (Chen et al., 2023a) collects human-annotated code refinement data to train a separate refinement model on it. Fhe refinement model is used to generate text-to-code data for finetuning the code-generation LLM. Unlike ILF, our approach is more scalable as we do not require human annotation and our personalized data proves significantly more effective than ILF as we empirically investigate in §5.

## 2.3 Reinforcement Learning from (Human) Feedback

After the launch of ChatGPT, aligning LLMs to human preference has drawn tremendous attention to research communities. As one of the most influential approaches in this direction, reinforcement learning from human feedback (RLHF) (Ouyang et al., 2022; Li et al., 2023b), adopts an actor-critic framework, where the student model is optimized to generate responses to receive higher reward from the critic model. In InstructGPT (Ouyang et al., 2022), the critic (reward model) is trained from human annotation. Direct Preference Optimization (DPO) (Rafailov et al., 2023) drops the need of training a reward model, by using a reference LLM and offline trajectories to estimate the reward. Chain-of-Hindsight (Liu et al., 2023) converts human preference annotations into simple natural language feedback, and thus turns RL optimization to conditional generation. In above methods, the

assumption is that there are no ground truth targets and thus they try to improve the LLM based on the assessment (critic) of multiple generated outputs. However, such RL-style training will be less effective and efficient to supervised finetuning, especially for challenging tasks with sparse rewards – e.g. sovling math puzzles or coding tasks. Unlike these methods, our approach can acquire "ground truth" outputs from a personalised teacher, thus supervised finetuning can be applied which makes the learning effective and efficient, even for challenging tasks like solving coding problems.

## 3 Method

### 3.1 Standard Distillation

Assume a dataset of code generation tasks $\mathcal{D} = \{(t, u)\}$ where each problem (or task) consists of a task instruction $t$ and a unit test collection $u$. During training, we have access to a teacher model $\pi_\phi$ and a student model $\pi_\theta$. The objective is to distill how the teacher solves code generation tasks to the student model, in the context of $\mathcal{D}$. For each task $(t, u)$, we first query the teacher $\pi_\phi(t)$ with the task instruction, to get a direct generated code snippet $c_\phi$. Then, we execute the generated code $c_\phi$ against unit test cases $u$ and get its execution feedback $f \leftarrow \text{EXEC}(c_\phi, u)$, where the EXEC function returns passed if the code passes all the unit tests, otherwise it returns an error message from the executor. By filtering out the tasks where $c_\phi$ do not pass all the unit tests (i.e., $f \neq$ passed), we get a new clean dataset $\mathcal{D}_{\text{STAND}} = \{(t, u, c)\}$, where each task consists a task instruction $t$, a suite of unit tests $u$ and a correct solution code $c$.

We then finetune the student model $\pi_\theta$ on $\{(u, c)\} \sim \mathcal{D}_{\text{STAND}}$, where the input is the task instruction $u$ and the output is the corresponding code solution $c$. We name this approach **STAND**.

### 3.2 Personalised Distillation

The STAND approach simply samples training examples (instructions and labels) from the prior distribution of the teacher model and feeds it to the student without considering the conditions of the student model. Inspired by modern education principles which advocates interactive and personalised learning experience, we propose personalised distillation: adapting teaching materials to student's current knowledge

**Algorithm 1** personalised distillation for code generation (PERsD-combined).

1: **Input:** Dataset $\mathcal{D}_{\text{STAND}}$, student LLM $\pi_\theta$, unit test executor EXEC, refinement template $T_{\text{refine}}$, teacher LLM $\pi_\phi$
2: $\mathcal{D}_{\text{refine}} \leftarrow \{\}$ ▷ *refinement data for finetuning*
3: $\mathcal{D}_{\text{code}} \leftarrow \{\}$ ▷ *direct generation data*
4: **for** $(t, u, c) \in \mathcal{D}_{\text{STAND}}$ **do**
5:     $c_\theta \leftarrow \pi_\theta(t)$          ▷ *student generates $c_\theta$*
6:     $f \leftarrow \text{EXEC}(c_\theta, u)$    ▷ *exec. feedback for $c_\theta$*
7:     **if** $f \neq$ `passed` **then**
8:         // *personalised refinement from teacher*
9:         $c_{\text{refine}} \leftarrow \pi_\phi(t, c_\theta, f)$
10:        // *create refinement task instruction*
11:         $t_{\text{refine}} \leftarrow T_{\text{refine}}(t, c_\theta, f)$
12:         **if** $\text{EXEC}(c_{\text{refine}}, u) =$ `passed` **then**
13:            $\mathcal{D}_{\text{refine}}.\text{insert}(\{t_{\text{refine}}, c_{\text{refine}}\})$
14:            $\mathcal{D}_{\text{code}}.\text{insert}(\{t, c\})$
15:         **end if**
16:     **end if**
17: **end for**
18: $\pi_{\theta*} \leftarrow \text{FINETUNE}(\pi_\theta, \mathcal{D}_{\text{refine}} + \mathcal{D}_{\text{code}})$

and capacity. We propose three variants:

**PERsD-combined** Algorithm 1 shows detailed steps for PERsD-combined. This method takes the standard distillation dataset $\mathcal{D}_{\text{STAND}}$ from §3.1 and first lets the student generate solutions for each task. Then it filters out the tasks where the student model can already solve correctly. For the remaining tasks, it obtains the teacher's personalised refinement conditioned on the student's attempt and its execution error feedback, and only keeps the tasks where the teacher's refinement is valid (i.e., passes all the unit test cases). Figure 1 visualizes these three steps.

For this final task-set, we create two datasets: i) $\mathcal{D}_{\text{code}}$ containing task instruction as input and teacher's direct answer as output, and ii) $\mathcal{D}_{\text{refine}}$ containing task refinement instruction as input and personalised refinement answer as output. The task refinement instruction (line 9 in Algorithm 1) is created by concatenating task instruction $t$, student's attempt $c_\theta$ and its execution feedback $f$ with a refinement template $T_{\text{refine}}$ (More details in Appendix C). Such refinement instruction turns standard code generation into a code refinement task, teaching the student how to refine its own solution. PERsD-combined then finetunes the student model on $\mathcal{D}_{\text{refine}}$ combined with $\mathcal{D}_{\text{code}}$.

**PERsD-refine** Similar to PERsD-combined, this variant follows line 1-15 of Algorithm 1 to collect refinement data $\mathcal{D}_{\text{refine}}$. However, it differs from the above model as it only uses $\mathcal{D}_{\text{refine}}$ to finetune the student model.

**PERsD** This variant takes the training data $\mathcal{D}_{\text{refine}}$ from PERsD-refine and replace the input of each data point from code refinement prompt to original task instruction. It thus trains the student model with personalised labels on code generation.

To illustrate the difference between personalised refinement and teacher's direct solution, we show a real example in Figure 2. The top shows the personalised refinement for the given task, while the bottom section shows the direct teacher's generation for the same task. Note how the teacher's direct generation is significantly different from the student model's attempt, while the teacher's refinement follows the student's attempt and improves upon it. We hypothesize that such adaptive refinement where the teacher aligns to student's generation, helps the student to learn more efficiently and effectively, similar to how humans benefit from personalised learning.

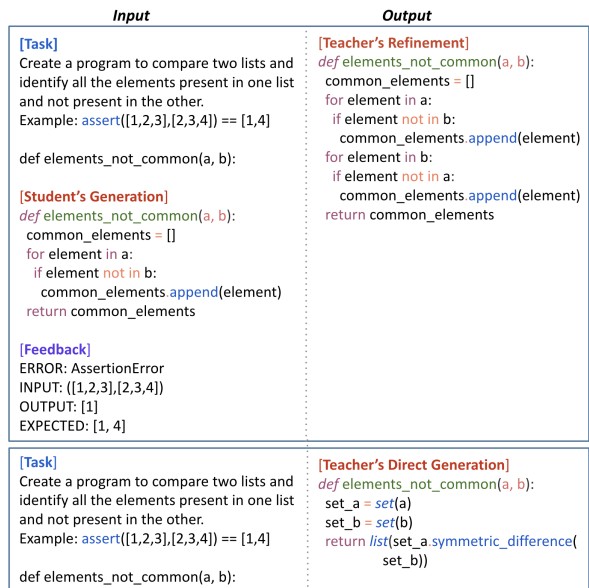

Figure 2: Example: (Top) Personalised refinement from student's attempt and execution feedback; (Bottom) Direct solution generated by teacher conditioned on task.

### 3.3 Iterative Inference

Let $\mathcal{D}_{\text{test}} = \{(t, u)\}$ denote our test set for inference, where each data point $(t, u)$ consists of a task instruction $t$ and a suite of hidden unit test cases $u$. We also assume that the task instruction contains some simple unit test cases in its doc-string (as often seen in code generation instructions), which we can extract and format using rule-based heuristics to obtain a suite of seen unit test cases $u_{\text{seen}}$ (More details in Appendix A).

For single-step inference, we use the standard approach to evaluate pass@k. Specifically, for each task $t$, we query the model $n$ times with the task instruction: $c_\theta^i \leftarrow \pi_\theta(t)$ for $i = 1 \ldots n$. Then, following (Chen et al., 2021), we estimate pass@k from the number of attempts that passed the hidden unit test cases: $\text{EXEC}(c_\theta^i, u) = \texttt{passed}$.

**Multi-step inference** If the model $\pi_\theta$ has been trained to rectify, following our approach in PERsD-refine or PERsD-combine, and if unit tests are available during inference, we can perform 2-step inference: for each generated attempt $c_\theta^i$ in 1-step, we first get execution feedback $f_{\text{seen}}^i \leftarrow \text{EXEC}(c_\theta^i, u_{\text{seen}})$. If $f_{\text{seen}}^i = \texttt{passed}$, we reuse the original attempt as the 2-step attempt. Otherwise, we create a refinement instruction $t^i \leftarrow T_{\text{refine}}(t, c_\theta^i, f_{\text{seen}}^i)$ following the approach in PERsD-refine or PERsD-combined, and query the same model with the refinement instruction for 2-step attempt: $c_{\theta,\text{2-step}}^i \leftarrow \pi_\theta(t^i)$. We then compute pass@k over the 2-step generations similar to 1-step inference.

# 4 Experimental Setup

## 4.1 Baselines

The first baseline is **STAND**, the standard distillation approach mentioned in §3.1.

To measure the effectiveness of personalised labels quantitatively, we also compare with **Input-personalised distillation** baselines as well, where only the input tasks are selected in a manner customized to the student's abilities. However, the output labels are not personalised, as they are taken from teacher's direction generation $c$ instead of personalised refinement $c_{\text{refine}}$. We start with $\mathcal{D}_{\text{code}}$ from PERsD-combined and have three variants:

**INPD** We finetune the student model $\pi_\theta$ on $\{(t, c)\} \sim \mathcal{D}_{\text{code}}$, where the input is a task instruction and the output is a code solution. This variant is more customized than STAND as it filters out the tasks which the student can already solve correctly.

**INPD-refine** Similar to PERsD-refine, InpD-refine trains the student model to rectify its wrong attempt. The difference is in InpD-refine, the refined code is from teacher's direct solution $c$, instead of personalised refinement $c_{\text{refine}}$.

**INPD-combined** Similar to PERsD-combined, InpD-combined trains the student on rectifying its

answers as well as directly solving the task. The difference is that in InpD-combined, the labels for both code refinement and code generation are taken from teacher's direct solution $c$.

## 4.2 Pretraining Data Construction

To construct our pretraining data, we adopted the data collection process in code-alpaca(Chaudhary, 2023) and used a set of 374 seed tasks from MBPP (task-ids 601-974) as in-context prompt to query ChatGPT for novel code generation tasks. This seed-set increases the likelihood of ChatGPT generating python codes.

Through this process, we obtained a corpus of 20K code generation tasks from ChatGPT each comprising a task instruction and the corresponding generated code, which is typically a single python function. Next we show each generated instance to ChatGPT again and prompt it to generate 5 unique test-case inputs (i.e. input argument values) for the python function. We then parse and format the generated test-case input and execute the generated code on it obtain an output. Thus, out of 20K, for 14880 instances we could successfully generate and parse 5 unit test case inputs and for 10172 instances we were able to successfully execute the generated code and obtain outputs on all 5 inputs. This final corpus of 10K code generation tasks, each comprising a task instruction and the corresponding generated code along with 5 unit test input and outputs forms our standard distillation dataset $\mathcal{D}_{\text{STAND}}$.

To collect personalised distillation data, we follow §3.2 to first ask the student model to generate 1 output code per task, setting sampling temperature to 0.3. We then evaluate the student's attempt and only keep the tasks with the wrong generations (i.e. the ones which failed any of the unit test-case). We use this to query ChatGPT for personalised refinements and only retain the valid refinements which passed all unit tests. Our prompt to ChatGPT contains the original task instruction and code from $\mathcal{D}_{\text{STAND}}$ along with the student model's generated code and execution feedback (compiler errors or unit test failures). Our instruction to ChatGPT is to generate a correct solution that rectifies the errors and is closest in semantics to the student's code (More details in Appendix B). Table 3 shows the statistics of personalised data construction process.

| Student Model | # Wrong Attempt by Student | # Validated Personalised Tasks | Data Cost |
|---|---|---|---|
| CodeGen-mono-6B (Nijkamp et al., 2023) | 6.5K | 3.25K | 5.5$ |
| CodeGen-mono-6B (round2) | 4K | 1.4K | 4.4$ |
| CodeGen-mono-16B | 6.2K | 2.8K | 6.5$ |
| StarCoder (Li et al., 2023a) | 4.3K | 2.5K | 4.3$ |

Table 3: Statistics of Personalised Data Construction

## 4.3 Model Evaluation

We evaluate our models on two datasets: HumanEval(Chen et al., 2021), which contains 164 Python problems, and the subset MBPP(Austin et al., 2021) sanitized set that has no overlap with our MBPP seed tasks for pretraining data collection. This corresponds to test+validation+prompt splits of MBPP-sanitized and consists of 306 Python problems. We use nucleus sampling with temperature 0.2 to generate 20 candidates per task for estimating pass@1, and with temperature 0.8, 100 candidates per task for estimating pass@5/10/20/50/100.

For multi-step inference, we first extract the "seen" unit test-cases from the doc-string of the task instruction (More details in Appendix A). Next, we generate output samples in the usual code-generation style forming the set of 1-step generations for each instance. Each of these candidate generations are then executed on the extracted "seen" unit test cases to obtain a refined code, thus forming the set of 2-step generations.

## 4.4 Pretraining Setup

For all experiments with CodeGen-mono-6B backbone, we use effective batch size of 1024 and pretrain for 20 epochs. For backbone as CodeGen-mono-16B, we use effective batch size of 1024 and pretrain for 3 epochs, as the training converges much faster than CodeGen-mono-6B. For PERsD-combine with StarCoder model, we use effective batch size of 1024 and pretrain for 8 epochs, which results in similar training loss as CodeGen-mono-16B. We implement using HuggingFace transformers(Wolf et al., 2020) and DeepSpeed Zero (Rajbhandari et al., 2020). All experiments are conducted on a cluster of 8 A100-40GB GPUs.

## 5 Experimental Results
## 5.1 Main Results

We empirically test the hypothesis that personalised distillation helps student model learn more effectively, by comparing PERsD models with baseline distillation methods (InpD, StanD) in Table 4.

**Personalised labeled-data is generally better than standard data** Comparing PERsD-combine to InpD-combine, we find PERsD-combine outperforms InpD-combine in all settings, often with a significant margin (two backbones, two datasets, two inference steps, 4 pass@k metric). Similar observation holds true when comparing PERsD-refine to InpD-refine (except for 2/32 settings), and PERsD to InpD. Thus, we conclude that PERsD-variants are generally significantly better than their InpD counterparts, providing strong evidence that personalised labels are more effective for the student model to learn than standard labels.

**PERsD outperforms StanD with less than one-third of its data** We observe that PERsD outperforms StanD for every pass@k on both 16B and 6B CodeGen-mono backbone across both HumanEval and MBPP, even though StanD has 10K data and PERsD has only 3.3K and 2.8K examples for CodeGen-mono-6B and 16B. The only exception is in the setting CodeGen-mono-16B, MBPP, pass@1, where StanD edges out PERsD by 1.2 points. Given that our pretraining data is constructed from seed tasks taken from MBPP, we hypothesize that StanD might enjoy an unfair advantage due to its having three times more data, making it more susceptible to data leakage. We verify such hypothesis further in §5.2. In summary, with PERsD outperforming StanD in 15 out of 16 settings while having less than a third of the data, it's evident that personalized labeled data makes the learning more efficient.

**Multi-step inference consistently improves answer quality** For PERsD-refine and PERsD-combine models, we find that 2 step inference consistently improves performance on HumanEval and MBPP. This shows the models successfully learn how to rectify its solution based on execution error feedback. Note that InpD-refine yields worse accuracy with 2 step inference on HumanEval pass@10/20, strengthening the advantage of personalised labeled data over standard labeled data.

## 5.2 Train-Test overlap analysis

As observed in Table 4, PersD-variants enjoy higher average improvements over their InpD counterparts, on HumanEvan than on MBPP. To delve deeper, we conduct a data overlap analysis. For each test task, we extract the most similar training task and use GPT-3.5-turbo to score their semantic similarity, with 0 indicating no relation and 1

(a) Backbone as CodeGen-mono-6B

| Methods | #Data | Pass@1 | | Pass@5 | | Pass@10 | | Pass@20 | |
|---|---|---|---|---|---|---|---|---|---|
| | | step=1 | step=2 | step=1 | step=2 | step=1 | step=2 | step=1 | step=2 |
| HumanEval | | | | | | | | | |
| StanD | 10K | 32.41 | - | 41.79 | - | 45.67 | - | 49.26 | - |
| InpD | 3.3K | 31.65 | - | 44.55 | - | 50.72 | - | 56.76 | - |
| -refine | 3.3K | 29.70 | 29.70 | 43.82 | 41.99 | 51.28 | 47.89 | 58.29 | 53.51 |
| -combined | 6.5K | 30.15 | 32.30 | 42.94 | 45.27 | 47.91 | 50.50 | 52.54 | 55.46 |
| PERsD | 3.3K | **34.63** | - | **49.34** | - | 55.34 | - | 60.41 | - |
| -refine | 3.3K | 32.35 | 33.35 | 48.69 | 49.35 | **56.07** | 56.87 | **63.60** | **64.76** |
| -combined | 6.5K | 33.81 | **35.53** | 44.64 | **49.67** | 49.96 | 55.67 | 55.23 | 61.21 |
| MBPP | | | | | | | | | |
| StanD | 10K | 43.11 | - | 55.24 | - | 59.07 | - | 62.51 | - |
| InpD | 3.3K | 43.59 | - | 55.83 | - | 63.13 | - | 67.34 | - |
| -refine | 3.3K | 44.44 | 47.81 | 62.25 | 66.43 | 67.61 | 71.44 | 71.68 | 75.22 |
| -combined | 6.5K | 42.69 | 47.25 | 56.70 | 62.17 | 61.39 | 66.49 | 65.46 | 70.22 |
| PERsD | 3.3K | 45.47 | - | 59.90 | - | 64.85 | - | 69.73 | - |
| -refine | 3.3K | **48.24** | **52.65** | **63.65** | **68.49** | **69.00** | **73.34** | **73.16** | **77.62** |
| -combined | 6.5K | 42.77 | 48.92 | 56.91 | 62.29 | 61.43 | 66.89 | 65.22 | 70.96 |

(b) Backbone as CodeGen-mono-16B

| Methods | #Data | Pass@1 | | Pass@5 | | Pass@10 | | Pass@20 | |
|---|---|---|---|---|---|---|---|---|---|
| | | step=1 | step=2 | step=1 | step=2 | step=1 | step=2 | step=1 | step=2 |
| HumanEval | | | | | | | | | |
| StanD | 10K | 33.96 | - | 50.56 | - | 57.69 | - | 63.82 | - |
| InpD | 2.8K | 36.68 | - | 49.51 | - | 53.85 | - | 57.47 | - |
| -refine | 2.8K | 30.55 | 31.28 | 48.40 | 48.13 | 55.00 | 54.52 | 61.31 | 60.62 |
| -combined | 5.6K | 34.66 | 36.49 | 50.65 | 53.89 | 56.75 | 60.07 | 62.78 | 65.85 |
| PERsD | 2.8K | **37.74** | - | **56.57** | - | **63.92** | - | **69.97** | - |
| -refine | 2.8K | 36.77 | **37.99** | 51.86 | 54.23 | 58.07 | 60.92 | 63.17 | 67.13 |
| -combined | 5.6K | 36.40 | 37.74 | 53.57 | **55.80** | 60.81 | 63.37 | 67.3 | **70.50** |
| MBPP | | | | | | | | | |
| StanD | 10K | 48.90 | - | 62.21 | - | 66.91 | - | 71.33 | - |
| InpD | 2.8K | 46.27 | - | 58.45 | - | 62.61 | - | 66.43 | - |
| -refine | 2.8K | 48.79 | 54.87 | **66.89** | 71.32 | **72.24** | 75.71 | 75.82 | 78.84 |
| -combined | 5.6K | 47.39 | 53.59 | 59.14 | 66.38 | 63.48 | 70.76 | 67.10 | 74.35 |
| PERsD | 2.8K | 47.68 | - | 65.80 | - | 71.56 | - | 76.02 | - |
| -refine | 2.8K | **51.50** | 56.21 | 66.82 | **71.86** | 72.06 | **76.78** | 76.03 | **80.42** |
| -combined | 5.6K | 51.44 | **56.44** | 66.45 | 71.31 | 71.64 | 76.43 | **76.04** | 80.20 |

Table 4: Comparing PERsD models to StanD & InpD

indicating complete semantic overlap (further details in Appendix D). Table 5 reveals more overlap in MBPP than HumanEval, and more overlap for StanD compared to PERsD. This overlap could be why StanD surpasses PERsD in the 1/16 setting (CodeGen-mono-16B, MBPP, pass@1), as StanD has an unfair advantage of having significantly more data leakage. In addition, if we test our methods on clean-MBPP where the leaked data points are removed, then PERsD becomes almost on-par with StanD in this specific setting while having larger margin over StanD on the rest 15/16 settings (from 4.8 points average margin to 5.9 points, more details at Appendix E). Altogether, this overlap analysis, coupled with results from cleaned MBPP, further underscores the advantages of personalized distillation.

## 5.3 Effect of mixing StanD and InpD data

Table 6 shows the ablation study on mixing standard distillation data to PERsD-refine and InpD-refine: while mixing standard data to InpD-refine improves its 1-step performance on MBPP and

| Method | Backbone | %("leak") | Similarity |
|---|---|---|---|
| | | HumanEval | |
| StanD | 6B,16B | 6.1% | 0.22 |
| PERsD | 6B | 3.6% | 0.18 |
| PERsD | 16B | 3.05% | 0.22 |
| | | MBPP | |
| StanD | 6B,16B | 18.24% | 0.40 |
| PERsD | 6B | 8.47% | 0.30 |
| PERsD | 16B | 7.49% | 0.30 |

Table 5: Train-Test Overlap Analysis. 6B/16B denotes CodeGen-mono-{6/16}B backbones. %("leak") denotes the percentage of test data that are semantically leaked in training data. 'Similarity' represents the average similarity score (range: 0 to 1; higher values indicate greater similarity)

roughly maintains its performance on other settings, mixing StanD data to PERsD-refine significantly deteriorate its performance (except pass@1 inf-step=2 on HumanEval). We conjecture that as StanD has much larger data volume than PERsD-refine, it overwhelms the student training on standard distillation. However, combining with a balanced input-personalised data can be beneficial, as we observe from the good performance of PERsD-combined in Table 4 on CodeGen-mono-16B.

| Methods | Inf Step | Pass@1 | Pass@5 | Pass@10 | Pass@50 | Pass@100 |
|---|---|---|---|---|---|---|
| | | HumanEval | | | | |
| StanD + InpD-refine | 1 | 30.59 | 40.04 | 44.20 | 54.23 | 58.54 |
| StanD + InpD-refine* | | 29.45 | 39.83 | 44.07 | 54.55 | 59.76 |
| StanD + PERsD-refine | | 32.13 | 43.82 | 48.66 | 59.55 | 64.02 |
| PERsD-refine | | **32.35** | **48.69** | **56.07** | **72.10** | **77.44** |
| StanD + InpD-refine | 2 | 30.87 | 42.88 | 47.90 | 58.21 | 60.98 |
| StanD + InpD-refine* | | 30.12 | 42.71 | 47.42 | 58.69 | 64.02 |
| StanD + PERsD-refine | | **35.00** | 47.89 | 52.96 | 64.36 | 69.51 |
| PERsD-refine | | 33.35 | 49.35 | 56.87 | 74.13 | 79.88 |
| | | MBPP | | | | |
| StanD + InpD-refine | 1 | 42.60 | 53.18 | 56.49 | 62.11 | 63.07 |
| StanD + InpD-refine* | | 44.08 | 54.12 | 57.82 | 64.96 | 66.34 |
| StanD + PERsD-refine | | 45.63 | 53.20 | 56.38 | 63.02 | 65.36 |
| PERsD-refine | | **48.24** | **63.65** | **69.00** | **78.16** | **81.70** |
| StanD + InpD-refine | 2 | 46.32 | 58.84 | 62.80 | 69.80 | 71.23 |
| StanD + InpD-refine* | | 46.92 | 58.18 | 62.03 | 68.82 | 68.95 |
| StanD + PERsD-refine | | 48.44 | 58.37 | 62.47 | 70.64 | 73.20 |
| PERsD-refine | | **52.65** | **68.49** | **73.34** | **82.72** | **85.62** |

Table 6: Ablation on mixing StanD, with Backbone as CodeGen-mono 6B. InpD-refine* denotes using all 6.5K tasks where the student model made mistakes, which covers around 3K more tasks than InpD-refine.

Similarly, in Table 7 we show another ablation: that mixing InpD data with PERsD roughly maintains the performance on HumanEval but degrades on MBPP. This shows personalised labels are of higher quality and mixing non personalised labels for the same task generally hurts performance.

| Methods | Pass@1 | Pass@5 | Pass@10 | Pass@50 | Pass@100 |
|---|---|---|---|---|---|
| | | | HumanEval | | |
| PERsD | **34.63** | **49.34** | **55.34** | **65.56** | 67.93 |
| PERsD + InpD | 34.88 | 48.35 | 54.06 | 64.88 | 68.90 |
| | | | MBPP | | |
| PERsD | **45.47** | **59.90** | **64.85** | **76.05** | **80.07** |
| PERsD + InpD | 43.84 | 59.02 | 63.77 | 71.69 | 74.84 |

Table 7: Ablation on PERsD mixing InpD with CodeGen-mono 6B as backbone

| Round | Inf Step | Pass@1 | Pass@5 | Pass@10 | Pass@50 | Pass@100 |
|---|---|---|---|---|---|---|
| | | | | HumanEval | | |
| 1 | 1 | **33.81** | 44.64 | 49.96 | 61.75 | 70.73 |
| 2 | | 32.74 | **45.50** | **51.52** | **66.14** | **71.95** |
| 1 | 2 | 35.53 | 49.67 | 55.67 | 68.16 | **77.44** |
| 2 | | **36.75** | **49.71** | **56.13** | **70.24** | 75.00 |
| | | | | MBPP | | |
| 1 | 1 | 42.77 | 56.91 | 61.43 | 68.84 | 70.67 |
| 2 | | **45.07** | **57.75** | **62.27** | **70.49** | **72.55** |
| 1 | 2 | 48.92 | 62.29 | 66.89 | 75.09 | 77.25 |
| 2 | | **49.59** | **63.43** | **68.30** | **76.00** | **78.10** |

Table 8: Ablation on multi-round distillation on PERsD-combined with CodeGen-mono 6B as backbone

## 5.4 Multi-round Distillation

After finetuning the student model with the personalised distillation data, can we perform another round of personalised distillation, on the new model? We show such an ablation study in Table 8. Encouragingly, we find PERsD-combined round-2 generally outperforms PERsD-combined round-1 by a modest margin. This improvement provides further evidence of the benefits of personalized learning, even when applied to models trained with personalized distillation. These findings suggest the intriguing possibility of an online or active version of personalized distillation, where data collection and model training occur simultaneously to ensure each batch is fully personalized and has higher sample efficiency. However, we will leave such intriguing exploration for future work.

## 5.5 Utilizing feedback for multi-step Inference

To better understand the role of execution feedback during training and multi-step inference, we show an ablation study in Table 9, where we compare PERsD-combine with a specific variant (PERsD-combine*) that excludes feedback during both training and inference. we observed that PERsD-combine* performs comparably to PERsD-combine on HumanEval and slightly better on MBPP for 1-step inference. However, for 2-step inference, PERsD-combine* consistently underper-

forms PERsD-combine. This result aligns well with our expectations that code-rectification needs the execution feedback to guide the refinement.

| Methods | Inf Step | Pass@1 | Pass@5 | Pass@10 | Pass@50 | Pass@100 |
|---|---|---|---|---|---|---|
| | | | | HumanEval | | |
| PERsD-combine | 1 | **33.81** | 44.64 | 49.96 | 61.75 | **70.73** |
| PERsD-combine* | | 33.29 | **45.47** | **50.90** | **62.87** | 68.29 |
| PERsD-combine | 2 | **35.53** | **49.67** | **55.67** | **68.16** | **77.44** |
| PERsD-combine* | | 34.59 | 49.54 | 55.59 | 67.27 | 71.95 |
| | | | | MBPP | | |
| PERsD-combine | 1 | 42.77 | 56.91 | **61.43** | **68.84** | 70.67 |
| PERsD-combine* | | **44.76** | **56.95** | 60.85 | 68.67 | **71.57** |
| PERsD-combine | 2 | **48.92** | **62.29** | **66.89** | **75.09** | **77.25** |
| PERsD-combine* | | 47.83 | 61.28 | 65.54 | 73.03 | 75.49 |

Table 9: Ablation on removing execution feedback with CodeGen-mono 6B as backbone. PERsD-combine* denotes combined personalised distillation without execution feedback in input prompt.

## 5.6 Cross-Model Personalised Distillation

To investigate whether personalised distillation data of one model can be benefical to another, we conduct an ablation in Table 10 by using PERsD-combined data of CodeGen-mono-6B to train CodeGen-mono-16B. The results show that such cross-model personalised data do not perform as well as real personalised data: leading to a consistent performance drop by a large margin. This finding reinforces our notion that learning data should be tailored to the specific student model, as personalized data suitable for one model may not necessarily benefit others.

| Model | Inf Step | Pass@1 | Pass@5 | Pass@10 | Pass@50 | Pass@100 |
|---|---|---|---|---|---|---|
| | | | | HumanEval | | |
| CodeGen-mono-6B | 1 | 33.81 | 44.64 | 49.96 | 61.75 | 70.73 |
| CodeGen-mono-16B* | | 32.99 | 47.81 | 54.58 | 69.31 | 73.98 |
| CodeGen-mono-16B | | **36.40** | **53.57** | **60.81** | **74.64** | **79.88** |
| CodeGen-mono-6B | 2 | 35.53 | 49.67 | 55.67 | 68.16 | 77.44 |
| CodeGen-mono-16B* | | 35.85 | 51.31 | 58.23 | 74.02 | 76.60 |
| CodeGen-mono-16B | | **37.74** | **55.80** | **63.37** | **77.14** | **81.10** |
| | | | | MBPP | | |
| CodeGen-mono-6B | 1 | 42.77 | 56.91 | 61.43 | 68.84 | 70.67 |
| CodeGen-mono-16B* | | 43.24 | 60.14 | 65.19 | 72.31 | 74.19 |
| CodeGen-mono-16B | | **51.44** | **66.45** | **71.64** | **80.62** | **82.93** |
| CodeGen-mono-6B | 2 | 48.92 | 62.29 | 66.89 | 75.09 | 77.25 |
| CodeGen-mono-16B* | | 48.12 | 65.31 | 70.02 | 76.60 | 78.70 |
| CodeGen-mono-16B | | **56.44** | **71.31** | **76.43** | **84.39** | **86.76** |

Table 10: Ablation on cross-model personalised distillation with PERsD-combined. CodeGen-mono-16B* means distillation data is from CodeGen-mono-6B.

## 5.7 Comparison with other Feedback-based Code Generation Models

**Comparison with ILF** (Chen et al., 2023a): In order to compare with ILF, one of our closest related work, we experiment on a separate setting:

starting with full MBPP dataset (974 tasks) and use Task-Ids 11-111 as test split and remaining 863 as training data. On the training set, our student model CodeGen-6B (same as ILF) generated wrong attempts on 562 tasks, which were shown to ChatGPT along with the task instruction and execution error feedback to eventually collect 288 valid personalized code rectification labels.

The original MBPP text-to-code data and this collected personalized code-refinement data for the 288 tasks respectively form the finetuning data $\mathcal{D}_{\text{code}}$ and $\mathcal{D}_{\text{refine}}$ on which we train models PERsD and PERsD-refine. We further combine $\mathcal{D}_{\text{code}}$ and $\mathcal{D}_{\text{refine}}$ to train PERsD-combined.

| | MBPP Test Set | | |
|---|---|---|---|
| Method | Cost | Pass@1 | Pass@10 |
| ILF | >4K$ | 36 | **68** |
| PERsD | 0.65$ | 46.8 | 67.4 |
| -refine | 0.65$ | 41.8 | 66.8 |
| -combined | 0.65$ | **47.8** | 64.8 |

Table 11: Comparison with ILF

Our experimental results in Table 11 show that all PERsD-variants significantly outperform ILF by 11.8% at pass@1 at a cost 1e-4 times lower than ILF, thus showcasing the lack of scalability of ILF-style models.

**Comparison with Self-Edit**: Since Self-Edit (Zhang et al., 2023) uses a trainable CodeGen-350M code editor model and a frozen code-generation model, our experimental setup is not directly comparable with theirs. However, our INPD-refine and INPD-combined models can actually be considered as very close counterparts to a version of Self-Edit with shared a code-generation and code-refinement model and CodeGen-6B backbone. The consistent performance improvement of the personalized distillation models over the input-distilled ones across the board, alludes towards the prospect that PERsD-models are indeed more effective than Self-Edit style models.

### 5.8 Comparison with SOTA Models

Fianlly, we compare PERsD-combine models with open-source and close-sourced state-of-the-art models on HumanEval in Table 12. We find that PERsD-combine methods can significantly improve the backbone model, with a performance gain of 6.2 points for CodeGen-mono 6B (8.4% error reduction), 5.9 points for CodeGen-mono 16B (8.3% error reduction) and 12.2 points for Star-Coder (18.4% error reduction). Moreover, Star-Coder with PERsD-combined, outperforms other open-sourced models except WizardCoder. Note

| Model | Model size | Pass@1 | Pass@10 | Pass@100 |
|---|---|---|---|---|
| Closed-source models | | | | |
| LaMDA | 137B | 14.0 | - | 47.3 |
| PaLM | 540B | 26.2 | - | 76.2 |
| Codex | 12B | 28.8 | 46.8 | 72.3 |
| code-cushman-001 | - | 33.5 | 54.3 | 77.4 |
| code-davinci-002 | - | 47.0 | **74.9** | **92.1** |
| GPT-3.5 | - | 48.1 | - | - |
| phi-1 | 1.3B | 50.6 | - | - |
| GPT-4 | - | **67.0** | - | - |
| Open-source models | | | | |
| CodeGeeX | 13B | 22.9 | 39.6 | 60.9 |
| LLaMA | 65B | 23.7 | - | 79.3 |
| StarCoder | 15B | 33.6 | - | - |
| CodeGen-mono | 6B | 26.1 | 42.3 | 65.8 |
| CodeGen-mono | 16B | 29.3 | 49.9 | 75.0 |
| InstructCodeT5+ | 16B | 35.0 | 54.5 | 77.9 |
| WizardCoder | 15B | **57.3** | - | - |
| CodeGen-mono (PERsD-combined) | 6B | 33.8 | 50.0 | 70.7 |
| CodeGen-mono (PERsD-combined) | 16B | 36.4 | 60.8 | 79.9 |
| StarCoder (PERsD-combined) | 15B | 45.8 | **68.3** | **82.3** |

Table 12: Results of *pass@k(%)* on HumanEval

that our model ues 5K data examples while WizardCoder uses 78K. As mentioned in §2.1, WizardCoder is an orthogonal approach that can be integrated into personalised distillation.

## 6 Conclusion

In this paper, we introduced personalized distillation as a method for collecting customized labeled data that adapts to the capacity of student models, resulting in more effective learning. We demonstrated the advantages of personalized distillation over standard distillation in the field of code generation, achieving superior performance on both the HumanEval and MBPP datasets. Through comprehensive ablation studies, we confirmed that personalized distillation leads to higher data quality, benefits from multi-round distillation, and enables models to leverage execution feedback for self-rectification. We believe personalized distillation represents an exciting step towards better distillation of closed-source LLMs to open-source models.

## Limitations

In this section, we discuss some limitations of this paper and future directions to make it more valuable:

**On Data Scale** For a fair comparison, we have conducted all experiments based on the same 10K $\mathcal{D}_{\text{STAND}}$ data (introduced §4.2) and the corresponding personalised data processed from $\mathcal{D}_{\text{STAND}}$ are of size 2-3K as shown in Table 3. However, as we have proven personalised distillation supports

more effective and efficient learning, it is intriguing to investigate how well does personalised distillation scale with the data size. For example, if we scale personalised distillation data to 50K, how much more performance gain will PERsD methods receive compared to InpD and StanD with the scaling of data size.

**Online Personalised Distillation**   As discussed in §5.4, conducting a second round personalised distillation continues to improve a student model that is already trained with PERsD-combine. Such observation suggests the potential of an online version of personalised distillation, which collects a batch of personalised data on-the-fly with the teacher model, after each optimization step during finetuning. As we have proven that true personalised data is more beneficial than standard data or cross-model personalised data (§5.6), such online personalised distillation will in-principle maximally benefit from personalised distillation, as each batch of training data is fully tailored to the student model.

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

## A Details in Multi-step Model Evaluation

As the docstrings are ill-formated in HumanEval, we write a simple rule-based parsing code snippet to extract its seen unit test cases. On average per task, there is 2 seen unit test cases and 4.2 unseen unit test cases. The overlap between seen and unseen tests is 11.33%. For MBPP, since conventionally the instruction prompt is constructed by taking the task description and example usages (from the unit test cases) as part of the doc-string, we consider all the unit test cases to be "seen" and use all of them for multi-step inference.

## B ChatGPT Prompt Template for Personalised Distillation

In Figure 3, we show the prompt template we use to query ChatGPT for personalised refinement. For each task example, with task instruction $t$, unit test cases $u$ and correct code $c$, we query ChatGPT API with two turn conversation history.

For the first turn, we use the template in Figure 3a and replace «TASK», «HEADER» with the actual task instruction $t$ and function header extracted. This is added to first turn's user input and the correct code $c$ is included as first turn's assistant output. For the second turn, we use the template in Figure 3b and replace «CODE», «ERROR» with the student model's attempt and its execution feedback. This is added to second turn's user input and we query ChatGPT with the constructed converstaion history to get second turn's assistant output as personalised code refinement.

```
Complete the python function to solve the following task.
The input format is

TASK: << Some Task description >>
HEADER: << Header of the python function definition. Your
output should start with this header >>

Example

TASK:
<<TASK>>

HEADER:
<<HEADER>>

CORRECT CODE:
```

(a) Turn-1 Prompt Template

```
Now act like a python expert, whose objective is to rectify a
given incorrect code for the above task, based on the error
obtained in unit testing.

INCORRECT CODE:
<>

ERROR:
<<ERROR>>

Instructions for Code Rectification:
- Based on the above errors, rectify the logic of the
incorrect code and write a complete python function to
correctly solve the task.
- If the code is significantly wrong then you can completely
re-write the code, otherwise try to come up with a correct
version that is closest to the given code.
- All output code should be inside code block ``` ``` and start
with the given header.

RECTIFIED CODE:
```

(b) Turn-2 Prompt Template

Figure 3: Prompt templates to query personalised refinement. Top(a): prompt template for first turn conversation, Botton(b): prompt template for second turn conversation.

## C   Prompt Template for Code Refinement Finetuning

Figure 4 shows the refinement template $T_{\text{refine}}$ introduced in §3.2), which is used to construct input prompt for code refinement finetuning. we replace «TASK» with task instruction, «CODE» with the initial wrong attempt from student, «ERROR» with the execution feedback, and «HEADER» with function header extracted from task instruciton.

## D   Details in Data Overlap Analysis

This section describes the detailed procedures to conduct train-test data overlap analysis. The objective is to assess the extent of data leakage in the test datasets originating from our self-constructed pretraining corpus.

```
Rectify the below code for the given task based on
the errors obtained in the unit tests.

TASK:
<<TASK>>
<>

ERROR:
<<ERROR>>

<<TASK>>
<<HEADER>>
```

Figure 4: Prompt template for code refinement finetuning.

Firstly, we have performed exact string match and found no data leakage in any test data (HumanEval/MBPP).

To measure the semantic similarity between training/test tasks, we did the following:

1. For each task in the test (MBPP/HumanEval) we retrieve two closest training tasks (based on cosine similarity of starcoder embedding & tf-idf vectors of task description).

2. We use gpt-3.5-turbo-16k to identify whether there is a data leak between a train and test instance by classifying the pair into ("leak", "somewhat similar", "somewhat not similar", "not related"). We use a prompt with instructions and manually created few-shot examples and ask gpt-3.5 to generate the reasoning and categorization. We manually examined several examples per category to ensure the reasoning and judgment is done correctly and consistently.

3. Map the similarity categories to 0-1 similarity-score ("leak" -> 1, "somewhat similar" -> 0.75, "somewhat not similar" -> 0.25, "not related" -> 0) and show the mean score and % of cases classified as "leak". Note that StanD & PERsD have 10K & 3K training data respectively so their scores are different.

## E   Results in MBPP-Cleaned

In Appendix D, we find 55 data instances that are potentially leaked (with similarity score = 1) in MBPP test data. In this section, we construct a new MBPP-Cleaned dataset, where the leaked data points are removed (originally 306 problems → 251 problems after filtering). The results on this new MBPP-Cleaned dataset is shown in Table 13. From

the results, we can see for setting CodeGen-mono-16B, pass@1, PERsD becomes almost on-par with StanD (from a gap of -1.21 to -0.17). For the rest of 15/16 settings on PERsD comparing with StanD, its average margin is increased from 4.8 points to 5.9 points. Besides, PERsD-refine on MBPP-Cleaned shows more consistent and sizable improvements over InpD-refine, with an average edge of +0.86 for 1 step inference, and +1.91 for two step inference. Overall, with overlapped test data removed, PERsD methods show even larger advantages compared to StanD or InpD methods.

(a) Backbone as CodeGen-mono-6B

| Methods | #Data | Pass@1 | | Pass@5 | | Pass@10 | | Pass@20 | |
|---|---|---|---|---|---|---|---|---|---|
| | | step=1 | step=2 | step=1 | step=2 | step=1 | step=2 | step=1 | step=2 |
| MBPP-Cleaned | | | | | | | | | |
| StanD | 10K | 37.51 | - | 50.89 | - | 55.15 | - | 58.87 | - |
| InpD | 3.3K | 38.80 | - | 53.91 | - | 58.47 | - | 62.73 | - |
| -refine | 3.3K | 37.58 | 42.95 | 57.65 | 62.29 | 63.52 | 67.79 | 67.92 | 71.96 |
| -combined | 6.5K | 38.11 | 43.01 | 52.69 | 58.32 | 57.36 | 62.75 | 61.19 | 66.18 |
| PERsD | 3.3K | 41.30 | - | 56.20 | - | 61.86 | - | 67.53 | - |
| -refine | 3.3K | **43.86** | **47.73** | **59.33** | **64.41** | **65.19** | **69.95** | **69.62** | **74.33** |
| -combined | 6.5K | 38.86 | 43.75 | 52.78 | 57.04 | 57.35 | 61.78 | 61.52 | 66.19 |

(b) Backbone as CodeGen-mono-16B

| Methods | #Data | Pass@1 | | Pass@5 | | Pass@10 | | Pass@20 | |
|---|---|---|---|---|---|---|---|---|---|
| | | step=1 | step=2 | step=1 | step=2 | step=1 | step=2 | step=1 | step=2 |
| MBPP-Cleaned | | | | | | | | | |
| StanD | 10K | 43.10 | - | 57.53 | - | 62.92 | - | 68.12 | - |
| InpD | 2.8K | 40.64 | - | 53.88 | - | 58.82 | - | 62.88 | - |
| -refine | 2.8K | 43.67 | 49.60 | 63.14 | 68.21 | 69.27 | 73.28 | 73.36 | 76.85 |
| -combined | 5.6K | 41.63 | 47.77 | 54.74 | 62.24 | 59.67 | 67.33 | 63.75 | 71.57 |
| PERsD | 2.8K | 42.93 | - | 62.40 | - | 68.90 | - | 74.10 | - |
| -refine | 2.8K | **47.73** | **52.63** | **63.62** | **69.21** | **69.84** | **75.17** | **74.90** | **79.69** |
| -combined | 5.6K | 46.33 | 51.67 | 63.46 | 68.65 | 69.49 | 74.26 | 74.53 | 78.83 |

Table 13: Comparing performance of PERsD models to StanD & InpD on MBPP-Cleaned