# OpenReview forum: "Personalized Distillation: Empowering Open-Sourced LLMs with Adaptive Learning for Code Generation"
_EMNLP/2023/Conference — EMNLP 2023 Main_

### Official Review · Reviewer_ksiw · 2023-08-01

**Typos Grammar Style And Presentation Improvements:** None
**Soundness:** 4

**Excitement:**

4: Strong: This paper deepens the understanding of some phenomenon or lowers the barriers to an existing research direction.

**Missing References:**

None

**Paper Topic And Main Contributions:**

This paper proposes a new distillation method: the teacher model will provide refinement results based on student's failed attempt, and then student will learn the teacher's refinement results.

Compared with standard distillation where teacher model directly gave answers to students, the authors claim that their method is more personalized for student models and hence can achieve better distillation performance.

**Questions For The Authors:**

It is fair to do multi-step inference in the experiments?
I feel that is like to have a second chance for author's method to solve the task.
If it failed in 1-step and make it in 2-step, does this count pass@1 or pass@2?

**Reasons To Accept:**

The idea is smart and sound to me.

From Table 4 (a), (b), the proposed method largely outperforms standard method to do distillation.

**Reasons To Reject:**

The idea seems a general way to do distillation from ChatGPT to public small models on any tasks.
But authors only focus on code generation in this paper, so I wonder the generalization ability of this method.



**Reproducibility:**

4: Could mostly reproduce the results, but there may be some variation because of sample variance or minor variations in their interpretation of the protocol or method.

**Reviewer Confidence:**

4: Quite sure. I tried to check the important points carefully. It's unlikely, though conceivable, that I missed something that should affect my ratings.

---

> ### Author Rebuttal · Authors · 2023-08-27
>
> > The idea seems a general way to do distillation from ChatGPT to public small models on any tasks. But authors only focus on code generation in this paper, so I wonder the generalization ability of this method.
>
> We sincerely appreciate the comment on “The idea seems a general way to do distillation”, which is exactly how we view our method. In ML perspective, standard distillation can be seen as forcing student’s prior to teacher’s prior, in spite of the fact that the two prior distributions could be very different. Personalized distillation, on the other hand, aims to adapt student’s prior to its improved self.
>
> In this paper, we propose PERsD idea and verify it comprehensively on the coding domain. The main reason we choose coding domain is that the answers can be objectively and automatically evaluated (whether or not they pass the unit tests), thus we can have a clear understanding of how well PERsD performs compared to StanD/InpD.
>
> As the first work in this direction, we hope it to elicit future works on verifying personalized distillation theoretically / applying it to more domains
>
>
> > It is fair to do multi-step inference in the experiments? I feel that is like to have a second chance for author's method to solve the task. If it failed in 1-step and make it in 2-step, does this count pass@1 or pass@2?
>
> Thanks for bringing up this important question! In all comparisons, we compare 1-step and 2-step separately. When inf-step is not mentioned explicitly (Table 10,11,6), only 1-step results are reported to ensure fairness. We will add a sentence in Section 4 to make this clear in the final version.

---

### Official Review · Reviewer_Y76m · 2023-08-05

**Soundness:** 3

**Excitement:**

3: Ambivalent: It has merits (e.g., it reports state-of-the-art results, the idea is nice), but there are key weaknesses (e.g., it describes incremental work), and it can significantly benefit from another round of revision. However, I won't object to accepting it if my co-reviewers champion it.

**Paper Topic And Main Contributions:**

This paper proposes a personalized distillation method to distill knowledge from close-source LLMs to smaller open-source LLMs. The student solves the tasks first, then the teacher provides an adaptive refinement to improve the student. The experiments show the proposed method outperforms traditional distillation methods with one third of the data.

**Reasons To Accept:**

- The paper is well written and is easy to follow.
- The experiments are comprehensive and solid.


**Reasons To Reject:**

- In my opinion, the idea of training with feedback is widely adopted in many research fields and seems not novel. Nowadays, LLMs also widely leverage human feedbacks with reinforcement learning to train a better model. What are the main contributions of this paper compared with previous methods and those human feedback-based training methods in other areas?

**Reproducibility:**

3: Could reproduce the results with some difficulty. The settings of parameters are underspecified or subjectively determined; the training/evaluation data are not widely available.

**Reviewer Confidence:**

2: Willing to defend my evaluation, but it is fairly likely that I missed some details, didn't understand some central points, or can't be sure about the novelty of the work.

---

> ### Author Rebuttal · Authors · 2023-08-27
>
> > How does this paper compare to training with human feedback and reinforcement learning
>
> We highly appreciate the comments from the reviewer! To clarify the differences and links with feedback-based training of LLMs, we will provide a short survey here:
>
> “The reinforcement learning from human feedback (RLHF) [1,2], adopts an actor-critic framework, where the student model is optimized to generate responses which can receive higher reward from the critic model. In InstructGPT[1], the critic (reward model) is trained from human annotation. Direct Preference Optimization (DPO) [3] drops the need of training a reward model, by using a reference LLM and offline trajectories to estimate the reward. Chain-of-Hindsight [4] converts human preference annotations into simple NL feedback, and turns RL optimization to conditional generation. “
>
> The methods above can all be broadly categorized as RLHF-style training, as they try to improve a LLM based on its generation and preference feedback. While Learning from Feedback is both in RL-style-training and personalized distillation, the role and objective is different.
>
> **RLHF-style training**: For RLHF-style training, the model needs to be already competent enough such that “generally” one of its generations should be correct/good, in order to learn effectively from scalar rewards. This issue will be exacerbated when the task is hard and the reward is sparse – for example, solving math puzzles or coding tasks.
>
> **Personalized Distillation**:
> Compared to RL-style-training from (human) feedback, the fundamental difference is PERsD gets the “correct” answer from the teacher instead of just a preference feedback signal. This allows effective and efficient knowledge acquisition. In this case, the agent does not need to be competent enough to begin with, and can effectively learn to solve complex tasks like code generation.
>
> Compared to Standard Distillation, in which the student’s prior is adapted to the teacher’s prior despite the fact that they might be very different, PERsD allows the learning to be personalized and adapts the student’s prior to its improved self. In our paper, we have shown such personalized learning enjoys consistent empirical advantages over standard distillation.
>
> With the above two comparisons, PERsD actually enjoys the best of two worlds. In RL literature, “On-policy Imitation Learning”[5,6] shares similar spirit as PERsD, where the method is both on-policy (personalized) and imitation learning (distillation). However, the settings of [5,6] are traditional RL settings with multi-step trajectories and a small action space. Our setting differs as code-generation has only one-step trajectory and a rich action space (code output). In this perspective, our paper can be viewed as the first work which realizes “On-policy Imitation Learning” for real-world generation application with modern LLMs.
>
> Comparison with ILF (related work): In Sec 5.6 we compare with ILF, a related work which uses human-labeled personalized feedback for finetuning the model. In Table 10, we show PERsD achieves a much better pass@1 at a much lower cost than ILF.
>
> We hope the above short survey and comparison resolves R2’s concern. We thank R2’s original comments again for helping us clarify with and draw a link to recent methods of training llms with feedback. We will consider turning the above comparison into a new section in related work to our final draft.
>
>
> [1] Long Ouyang, et al. (2022). Training language models to follow instructions with human feedback.
>
> [2] Zihao Li, Zhuoran Yang, & Mengdi Wang. (2023). Reinforcement Learning with Human Feedback: Learning Dynamic Choices via Pessimism.
>
> [3] Rafael Rafailov, Archit Sharma, Eric Mitchell, Stefano Ermon, Christopher D. Manning, & Chelsea Finn. (2023). Direct Preference Optimization: Your Language Model is Secretly a Reward Model.
>
> [4] Hao Liu, Carmelo Sferrazza, & Pieter Abbeel. (2023). Chain of Hindsight Aligns Language Models with Feedback.
>
> [5] Kwangjun Ahn, Zakaria Mhammedi, Horia Mania, Zhang-Wei Hong, Ali Jadbabaie, Model Predictive Control via On-Policy Imitation Learning (2022)
>
> [6] Stephane Ross, J. Andrew Bagnell, Efficient Reductions for Imitation Learning (2010)

---

### Official Review · Reviewer_cUGP · 2023-08-11

**Typos Grammar Style And Presentation Improvements:** 1. There seems to be an error in the …
**Soundness:** 3

**Excitement:**

3: Ambivalent: It has merits (e.g., it reports state-of-the-art results, the idea is nice), but there are key weaknesses (e.g., it describes incremental work), and it can significantly benefit from another round of revision. However, I won't object to accepting it if my co-reviewers champion it.

**Paper Topic And Main Contributions:**

This paper presents a novel distillation method in the field of code generation for extracting capabilities from large closed-source models into smaller open-source models, improving the ability of open-source models to solve fundamental programming problems.
1. While I found the paper to be minimally challenging and not technically challenging, the starting point and motivation for "personalized learning" is compelling, and the authors have done sufficient experiments to validate this motivation.
2. There has been relatively little research in the code area on distillation of closed-source macromodels, and this paper extends that direction.

**Questions For The Authors:**

1. Is the 2-step refinement instruction for Multi-step inference in Section 3.3 generated with the help of ChatGPT? Does this go against the previously mentioned reduction of dependency on closed-source macromodels?
2. How were the training data for Round 2 of Multi-round Distillation in Section 5.3 obtained? How is it different from the data of Round 1?
3. I found that the experimental results of training CodeGen-momo-16B with the data of CodeGen-momo-6B in Table 9 are lower than the results of StanD in Table 4(b), does it mean that a separate training data must be developed for each model in order for the results to be better than the results of StanD? Does this suggest that this method does not generalize well?
4. How was the Cost>4K$ figure for ILF in Table 10 obtained? I didn't find in the original article that it costs that much.

**Reasons To Accept:**

1. This paper presents Personalized Distillation in the field of code generation, which improves the ability of some open source code models to solve basic programming problems with a small amount of training data.
2.The writing style of this paper is very good, the author can make good use of diagrams and graphs to get the ideas and thoughts across, and the experimental design is very adequate.

**Reasons To Reject:**

1. Only basic programming problems (e.g., HumanEval and MBPP) were focused on and not evaluated above complex programming problems (e.g., APPS or CodeContest).
2. The training data in this paper is not publicly available, which is not conducive to the reproduction of the experimental results.
3. Section 5.7 of the paper mentions that personalized distillation is orthogonal to WizardCoder, but there is no experimental data to prove this.
4. Section 5.1 of the paper mentions that "Comparing PERsD-refine to InpD-refine, we find PERsD-refine outperforms InpD-refine in all settings (two backbones, two406datasets, two inference steps). two406datasets, two inference steps)." But I find that in the experimental results of Table 4(b) MBPP, only the pass@1 result of PERsD-refine is a sizable improvement compared to InpD-refine, and the other results don't seem to be improved. And the pass@1 results for PERsD seem to be lower than those for StanD, which is not well explained in the paper.
5. There does not seem to be any experimental data in the paper on whether the different training data generated by ChatGPT overlap with the data from the test benchmarks HumanEval and MBPP, and the experimental results are not very rigorous.

**Reproducibility:**

3: Could reproduce the results with some difficulty. The settings of parameters are underspecified or subjectively determined; the training/evaluation data are not widely available.

**Reviewer Confidence:**

5: Positive that my evaluation is correct. I read the paper very carefully and I am very familiar with related work.

---

> ### Author Rebuttal · Authors · 2023-08-27
>
> We appreciate that R1 finds the motivation for “personalized learning” compelling and validated by sufficient experiments. However, we disagree with the arguments that the paper is “minimally challenging and not technically challenging”. We hope to clarify the importance and challenges of our approach with some additional information here:
>
> > “Training data not publicly available”:
>
> As we mention in our paper, we will release all our model checkpoints, codes and training data upon acceptance.
>
> > “Minimally Challenging”:
>
> From ML perspective, standard distillation – both white-box and black-box (Section 2.1) – can be seen as forcing student’s prior to teacher’s prior, in spite of the fact that the two prior distributions could be very different. In our work, we explore a fundamentally different approach: adapting the student’s prior to its improved self, acquired by asking the teacher to correct the attempt such that the resulting distribution is still close to the student's original prior. We motivated such an approach from “personalized learning” in human education and validated it with comprehensive experiments.
>
> To the best of our knowledge, such a personalized distillation idea has not been studied in the literature and it is a timely topic as the teacher model needs to provide personalized refinements (a non-typical task requiring general instruction following ability), which is only possible with large LLMs nowadays.
>
> > Not technically challenging. Why is personalized distillation orthogonal to WizardCoder?
>
> Code generation on HumanEval is still challenging for open-source code-LLMs (StarCoder only achieving 33.6% pass@1). Our PERsD can boost the performance to 45.8% pass@1 with only ~3k data points. The contemporary work WizardCoder achieves the SOTA among open-source models with 78k data points.
> WizardCoder is an orthogonal approach, as it proposes to generate more complex/diverse task instructions via evolve-instruct, while PERsD can take the data points generated by WizardCoder, and apply personalized distillation upon it. However, since the WizardCoder-78k data is not open-sourced, we couldn’t integrate it with PERsD.
>
> > Why not evaluated on more complex programming problems (e.g., APPS or CodeContest)
>
> HumanEval and MBPP are still quite challenging to smaller open-source LLMs (6B-16B). For example, StarCoder and CodeT5+ – two best models (at the time of submission), can achieve less than 35% pass@1 on HumanEval. Hence we conduct comprehensive evaluation on them across multiple settings, and validate the power of personalized distillation.
>
> We are aware of the more challenging coding datasets like APPS, CodeContest, XCodeEval. However, preliminary results on challenging datasets like APPS shows that even WizardCoder can only achieve around 10% / 2% / 0.4% pass@1 on introductory / interview / competition level problems. As such, these competitive programming problems might require some additional effort like curriculum learning + self-critique/reasoning along with personalized distillation, which we will leave to future work.
>
> > Overlap of ChatGPT generated training data with HumanEval & MBPP
>
> We have done a rigorous inspection of the possible overlap, as follows:
>
>
> Firstly, no exact match was found in training and test task instructions.
>
> To measure the semantic similarity between training/test tasks, we did the following:
> 1. For each task in the test (MBPP/HumanEval) we retrieve two closest training tasks (based on cosine similarity of starcoder embedding & tf-idf vectors of task description).
> 2. We use gpt-3.5-turbo-16k to identify whether there is a data leak between a train and test instance by classifying the pair into (“leak”, “somewhat similar”, “somewhat not similar”, “not related”). We use a prompt with instructions and manually created few-shot examples and ask gpt-3.5 to generate the reasoning and categorization. We manually examined several examples per category to ensure the reasoning and judgment is done correctly and consistently.
> 3. Map the similarity categories to 0-1 similarity-score (“leak” -> 1,  “somewhat similar” -> 0.75, “somewhat not similar” -> 0.25, “not related” -> 0) and show the mean score and % of cases classified as “leak”.  Note that StanD & PERsD have 10K & 3K training data respectively so their scores are different.
>
> Results:
>
> —----------------—----------------—----------------—----------------
>
> Test: HumanEval
>
> | method | backbone         | %("leak") | mean similarty-score |
> |--------|------------------|-----------|----------------------|
> | StanD  | \                | 6.1%      | 0.22                 |
> | PERsD  | Codegen-mono-6B  | 3.6%      | 0.18                 |
> | PERsD  | Codegen-mono-16B | 3.05%     | 0.22                 |
>
>
> —----------------—----------------—----------------—----------------
>
> Test: MBPP
>
>
> | method | backbone         | %("leak") | mean similarty-score |
> |--------|------------------|-----------|----------------------|
> | StanD  | \                | 18.24%    | 0.40                 |
> | PERsD  | Codegen-mono-6B  | 8.47%     | 0.30                 |
> | PERsD  | Codegen-mono-16B | 7.49%     | 0.30                 |
>
> From the table, we can see
> 1. There is more overlap for MBPP than HumanEval, as is expected since a part of the MBPP data was sequestered away as seed for generating training data.
> 2. StanD has more overlap than PERsD (or InpD).
>
> To the best of our knowledge, we are (one of) the first group providing such semantic overlap analysis (unfortunately it’s not seen in Starcoder, CodeT5+, WizardCoder, etc). We suspect WizardCoder might have higher data leak, as they use pass@1 on humanEval as selection criteria of their generated Evol-instruct type finetuning data.
>
> We will add the above semantic data overlap analysis to our final draft and provide more details in the appendix.
>
> > For MBPP, only the pass@1 result of PERsD-refine is a sizable improvement compared to InpD-refine,Pass@1 results for PERsD seem to be lower than those for StanD
>
> This is definitely an important observation. We appreciate the Reviewer's attention to details and apologize for not properly addressing it in the paper. For better clarity, we first summarize the average improvements PERsD achieves over StanD and InpD in the different settings below.
>
> —----------------—----------------—----------------—----------------
>
> MBPP: Avg Improvement across P@1,5,10,20
>
> | method                      | Codegen-mono-6B | Codegen-mono-16B |
> |-----------------------------|-----------------|------------------|
> | PERsD vs StanD              | + 5.0           | + 2.93           |
> | PERsD vs InpD               | + 2.52          | + 6.83           |
> | PERsD-refine vs InpD-refine | + 2.02          | + 0.56           |
> | PERsD-comb vs InpD-comb     | + 0.02          | + 7.12           |
>
> —----------------—----------------—----------------—----------------
>
> HumanEval:  Avg Improvement of across P@1,5,10,20
>
> | method                      | Codegen-mono-6B | Codegen-mono-16B |
> |-----------------------------|-----------------|------------------|
> | PERsD vs StanD              | + 9.1           | + 6.63           |
> | PERsD vs InpD               | + 4.01          | + 7.67           |
> | PERsD-refine vs InpD-refine | + 4.41          | + 3.65           |
> | PERsD-comb vs InpD-comb     | + 2.53          | + 3.31           |
>
>
> We observe the advantage of personalized distillation is generally lesser on MBPP than on HumanEval, though the overall averages indicate it still leads to 2-7% gains in the majority of settings and performs at par in remaining two settings, without actually hurting performance.
>
> One of the main reasons behind this is the data-overlap of MBPP being higher than HumanEval. More specifically, StanD enjoys a much higher (almost x3) data-overlap in MBPP over HumanEval. Additionally, StanD training data is almost 3x that of PERsD models. This means StanD has an unfair advantage (more data and more related data) which explains why it is possible for it to slightly outperform PERsD in 1/16 settings. However PERsD still outperforms StanD over 15/16 settings, showing the value of our proposal.
>
> We will summarize and add the above interpretation to our final version.
>
>
>
> > Limited improvement of PERsD over InpD in -refine and -combined settings:
>
> Following our data-overlap analysis we want to emphasize that MBPP is more of an in-domain evaluation (as pretraining data constructed from seeds tasks of MBPP) and HumanEval is more of an out-of-domain (more general) evaluation. We find the effect of personalized distillation more pronounced when evaluating in “out-domain” setting (e.g. PERsD-refine has a clear edge over InpD-refine on HumanEval for all settings). In HumanEval the overlap-score of all the models being reasonably low, we see more regular patterns of average improvement across all settings in the above table. Thus our main evaluations and comparison with SoTA has been only on HumanEval.
>
>
> > Is 2-step refinement for Multi-step inference done using ChatGPT?
>
> No, all inference is done by the local llm. We uphold the value that research and working products should be based on open sourced llms. The iterative inference is just a natural property that the model achieves simply because of the inherent design of the personalized data collection. There being quite limited work on using small open LLM for iterative self-correcting code-generation, PERsD advances this research as well by performing better than them (Sec 5.6 of paper)
>
> However our main goal/experiments show that personalized data in-fact helps boost 1-step code-generation, while the code-refinement step helps further.
>
> > How is Round 2 training data in Multi-round Distillation obtained
>
> Round2 training data is generated by treating the model finetuned in Round1 as the student. Round1 uses the original off-the-shelf pretrained model.
>
> > I found that the experimental results of training CodeGen-momo-16B with the data of CodeGen-momo-6B in Table 9 are lower than the results of StanD in Table 4(b), does it mean that a separate training data must be developed for each model in order for the results to be better than the results of StanD? Does this suggest that this method does not generalize well?
>
>  “not generalize” is actually an expected property, as it is intuitive that personalized data must be truly personalized. This is infact verified by our experiments: Training 16B model with 6B model’s personalized data should perform poorer than standard distillation. The prior of 6B leads to inferior quality code-generation than what the more powerful pretrained 16B model can generate, which causes it to actually unlearn itself.
>
> In human education, personalized teaching also costs more effort, but it’s more effective and more data efficient. In addition, it is understandable that learning the proxy data generated by another less competent student, resulting in worse performance than learning from the teacher.
>
> > Evidence that Cost of ILF (related work) is >4k
>
> From ILF paper,  "The final dataset consists of 195 triples of (incorrect program, human written feedback, human-written refinement). On average, workers are paid 23 dollars per annotated sample and take 27 minutes/sample" -- Assuming only 1 worker per task, the cost comes to 4.4K dollars

---

### Meta-Review · Area_Chair_ndRn · 2023-09-19

**Recommendation:** 4

**Metareview:**

The manuscript introduces a personalized distillation method to extract knowledge from closed-source to open-source Large Language Models. Unlike conventional distillation that forces student’s prior closely to follow teacher’s prior, the student LLM tackles given tasks first, then the teacher LLM adaptively offers refinement to improve the student’s own prior. The experimental results show that the personalized distillation outperforms the conventional distillation with only one third of the data. All reviewers concur that the paper proposes a smart idea, and the experimental results are solid for code generation domain. To further improve the clarity, incorporating friendly comparisons against traditional distillation and RLHF elucidates a clearer distinction across various approaches.

---

### Decision · Program_Chairs · 2023-10-07

**Decision:**

Accept-Main

**Comment:**

The manuscript introduces a personalized distillation method to extract knowledge from closed-source to open-source Large Language Models. Unlike conventional distillation that forces student’s prior closely to follow teacher’s prior, the student LLM tackles given tasks first, then the teacher LLM adaptively offers refinement to improve the student’s own prior. The experimental results show that the personalized distillation outperforms the conventional distillation with only one third of the data. All reviewers concur that the paper proposes a smart idea, and the experimental results are solid for code generation domain. To further improve the clarity, incorporating friendly comparisons against traditional distillation and RLHF elucidates a clearer distinction across various approaches.